# Bitter Orange (*Citrus aurantium* Linné) Improves Obesity by Regulating Adipogenesis and Thermogenesis through AMPK Activation

**DOI:** 10.3390/nu11091988

**Published:** 2019-08-22

**Authors:** Jinbong Park, Hye-Lin Kim, Yunu Jung, Kwang Seok Ahn, Hyun Jeong Kwak, Jae-Young Um

**Affiliations:** 1College of Korean Medicine, Kyung Hee University, Seoul 02447, Korea; 2Basic Research Laboratory for Comorbidity Regulation, Comorbidity Research Institute, Kyung Hee University, Seoul 02447, Korea; 3Life Science Major, Division of Bio-Convergence, College of Convergence and Integrate Science, Kyonggi University, Suwon 16227, Korea

**Keywords:** bitter orange, obesity, adipogenesis, thermogenesis, AMP-activated protein kinase

## Abstract

Obesity is a global health threat. Herein, we evaluated the underlying mechanism of anti-obese features of bitter orange (*Citrus aurantium* Linné, CA). Eight-week-administration of CA in high fat diet-induced obese C57BL/6 mice resulted in a significant decrease of body weight, adipose tissue weight and serum cholesterol. In further in vitro studies, we observed decreased lipid droplets in CA-treated 3T3-L1 adipocytes. Suppressed peroxisome proliferator-activated receptor gamma (PPARγ) and CCAAT/enhancer binding protein alpha indicated CA-inhibited adipogenesis. Moreover, CA-treated primary cultured brown adipocytes displayed increased differentiation associated with elevation of thermogenic factors including uncoupling protein 1 and PPARγ coactivator 1 alpha as well. The effects of CA in both adipocytes were abolished in AMP-activated protein kinase alpha (AMPKα)-suppressed environments, suggesting the anti-adipogenic and pro-thermogenic actions of CA were dependent on AMPKα pathway. In conclusion, our results suggest CA as a potential anti-obese agent which regulates adipogenesis and thermogenesis via AMPKα.

## 1. Introduction

Obesity (body mass index, BMI > 30 kg/m^2^) has risen into a serious health threat worldwide, especially in developed countries [1]. According to an analytic study of 19.2 million participants from 1975 to 2014, age-standardized BMI increased from 21.7 to 24.2 kg/m^2^ in men and 22.1 to 24.4 kg/m^2^ in women globally. Speculations based on this trend expects obesity prevalence of 2025 as 18% in men and 21% in women worldwide [2]. This global trend in concerning, as obesity is associated with high risks of comorbidities such as type 2 diabetes mellitus [3], cardiovascular diseases [4] and even certain cancers [5].

Obesity is a complex status in which multiple factors participate. Above all, when energy consumption exceeds expenditure, the surplus energy is stored as lipid, specifically triglyceride, in white adipose tissues (WATs) [6]. While WAT works as a storage unit to save excessive energy, brown adipose tissue (BAT), on the other hand, functions in an opposite way. BAT is an organ which can dissipate energy as heat in response to cold exposure or other stimuli such as pharmacological agents [7]. This process, the non-shivering thermogenesis, is mediated by the mitochondrial uncoupling protein 1 (UCP1), which shunts the proton circuit and leads to heat production instead of ATP [8].

Adipogenesis is the differentiation process of which preadipocytes develop into a mature status associated with accumulation of lipid droplets. Excessive lipid accumulation in WAT is both the cause and main symptom of obesity [9]. Multiple processes regulate adipogenesis, including adipocyte proliferation and differentiation or fatty acid oxidation and synthesis, and are controlled by numerous factors [10]. Studies suggest peroxisome proliferator-activated receptor gamma (PPARγ) and CCAAT/enhancer binding protein alpha (C/EBPα) as the key regulators of this process [11,12]. Inhibiting these two factors appears to be an attractive strategy for obesity treatment, because excessive growth of WAT in obese individuals has been suggested to be caused from not only proliferation of adipocytes but also a result of adipocyte hypertrophy by increased lipid accumulation in newly matured adipocytes as a result of adipogenesis [13,14].

Besides reducing accumulated lipid within WAT, regulation of BAT differentiation and activation is another attractive target for obesity care. BAT, first identified in the 1960’s, is a tissue organ which can dissipate energy in the form of heat; non-shivering thermogenesis. Thermogenesis in BAT is controlled by the activation of sympathetic nervous system (SNS), which releases noradrenaline (NA) to activate the β3-adrenergic receptor (β3-AR) in brown adipocytes [15,16,17]. In response to the activation of β3-AR, mitochondrial UCP1 provides a proton leak into the existing potential difference in the mitochondrial membrane, shunts the ATP-synthesizing respiratory chain, and results in facilitation of heat production. In the perspective of enery expenditure (EE), the activation of BAT has gained interest as a new promising therapeutic target for obesity management.

Bitter orange (*Citrus aurantium* Linné, CA), of which Latin name is Aurantii Fructus Immaturus, is also called as Seville Orange, sour orange or marmalade orange, and has found as a beverage and a dietary supplement on the market. Recent studies report the hepatoprotective [18], anti-cancer [19,20], anxiolytic [21,22], anti-oxidant [23,24] effects of CA. Furthermore, a number of studies have elucidated its beneficial effect on metabolic diseases as well [25,26,27,28]. However, up to date, regardless of the studies reporting various functions of CA, there has been no study investigating the influence of CA on brown adipocyte differentiation and activation. Here, our study were aimed to examine the anti-adipogenic and thermogenic mechanisms of CA in 3T3-L1 adipocytes and primary brown adipocytes, and in addition, elucidate the role of AMP-activated protein kinase (AMPK) in the action of CA.

## 2. Materials and Methods

### 2.1. Preparation of CA

CA, also called as Jigak in Korea, was purchased from Omniherb, Co. (Daegu, Republic of Korea). A voucher specimen of CA has been deposited in the College of Korean Medicine, Kyung Hee University (Seoul, Republic of Korea). CA was extracted with 80% ethanol for 2 h 20 min in a heating mantle. The solvents were filtered, freeze-dried (Rotary evaporator Model NE-1 and Freeze dryer FD-1, Tokyo, Japan), and then stored at −80°C until usage. The yield (w/v) was 25%.

### 2.2. Chemical Reagents and Antibodies

3-isobutyl-1-methylxanthine (IBMX), compound C (CC), dexamethasone (Dex), indomethacin, insulin, and Oil Red O powder were purchased from Sigma-Aldrich Co. (St Louis, MO, USA). Bovine serum (BS), Dulbecco’s modified Eagle’s medium (DMEM), penicillin/streptomycin/glutamine (P/S/G) and were obtained from Gibco (Grand Island, NY, USA). Fetal bovine serum (FBS) was purchased from HyClone Laboratories Inc. (Logan, UT, USA).

Anti-C/EBPα, anti-glyceraldehyde-3-phosphate dehydrogenase (GAPDH), anti-PPARγ coactivator 1 alpha (PGC1α) and anti-UCP1 antibodies were purchased from Santa Cruz Biotechnology, Inc. (Santa Cruz, CA, USA), and antibodies for AMPKα, pAMPKα, acetyl-CoA carboxylase (ACC), pACC, liver kinase B1 (LKB1), pLKB1 and PPARγ were purchased from Cell Signaling Technology, Inc. (Beverly, MA, USA).

### 2.3. Chromatographic Separation

The high-performance liquid chromatography (HPLC) analysis was performed with a vacuum degasser, a quaternary pump and an automatic sample injection system. The column Nucleosil C18 (150 × 4.6 mm, 5 µm, Teknokroma, Barcelona, Spain) separated the sample, 80% ethanol extract of CA, as the mobile phase at a flow rate of 1.0 ml/min at 25 °C. Initial elution was performed by acetonitrile–aqueous ammonium acetate 35:65 (v/v). After 30 min, the linear gradient reached 60% acetonitrile.

### 2.4. Ethics Statement

All animal experiment procedures were approved by the Animal Care and Use Committee of the Institutional Review Board of the Kyung Hee University (confirmation number: KHUASP (SE)-13-012).

### 2.5. Animals and Diets

Four-week-old male C57BL/6J mice (Daehan Biolink Co.,Eumsung, Korea) were maintained 1 week prior to the experiments for acclimatization. The mice were fed with a 60% kcal high fat diet (HFD) (Rodent diet D12492, Research diet, New Brunswick, NJ, USA) for 4 weeks to induce obesity in accordance with our previous reports [29,30,31]. Then, mice were divided into two groups (*n* = 5), fed for eight additional weeks with either a) HFD and b) HFD plus CA (100 mg/kg/day). The administration dose of CA was decided based on a previous study [32]. A group fed normal chow diet (ND) for twelve weeks were used as normal control. Body weight was measured two times per week. The composition of each diet is displayed in Appendix A.

### 2.6. Serum Analysis

The serum was separated by centrifugation (4000× *g*, 30 min) immediately after blood harvest via cardiac puncture. High-density lipoprotein (HDL) cholesterol, low-density lipoprotein (LDL) cholesterol and total cholesterol (TC) were assayed using a colorimetric enzyme-linked immunosorbent assay (ELISA) method.

### 2.7. Cell Culture and Differentiation

Murine 3T3-L1 preadipocytes were purchased from the American Type Culture Collection (Rockville, MD, USA), cultured in 10% BS/DMEM at 37 °C, 5% CO_2_ and differentiated into mature white adipocytes in 10% FBS/DMEM plus 0.5 mM IBMX, 1 μM Dex and 1 μg/mL insulin at 37 °C, 5% CO_2_ as previously reported [29].

Brown adipocytes were prepared as previously described [31]. Briefly, preadipocytes were isolated from interscapular BAT of new born (post-natal day 1-2) FVB/NJ mice (Daehan Biolink Co.,Eumsung, Korea), cultured in 10% BS/DMEM at 37 °C, 5% CO_2_ and differentiated into mature brown adipocytes in 10% FBS/DMEM plus 0.5 mM IBMX, 0.5 µM Dex, 20 nM insulin, 125 µM indomethacin and 1 nM T3 at 37 °C, 5% CO_2_.

### 2.8. Cytotoxicity Measurement

Cells were seeded (2 × 10^4^ cell/well) on 96 well plates, stabilized for 24 h, and incubated with various concentrations of CA (10-1000 μg/mL) for additional 48 h. Cell viability was monitored using the cell proliferation 3-(4,5-dimethylthiazol-2-yl)-5-(3-carboxymethoxyphenyl)-2-(4-sulfophenyl)-2H-tetrazolium (MTS) kit (Promega Co., Madison, WI, USA) according to the manufacturer’s instructions. The absorbance was measured at 490 nm with a VERSAmax microplate reader (Molecular Devices, Sunnyvale, CA, USA).

### 2.9. Oil Red O Staining

Intracellular lipid accumulation was measured by an Oil Red O staining assay as described previously [29]. The absorbance was measured at 500 nm in a VERSAmax microplate reader (Molecular Devices, Sunnyvale, CA, USA).

### 2.10. RNA Extraction and Real-Time Reverse Transcription Polymerase Chain Reaction (RT-PCR)

RNA was extracted using a GeneAllR RiboEx total RNA extraction kit (GeneAll Biotechnology, Seoul, Republic of Korea), cDNA reverse-transcription was performed using a Power cDNA synthesis kit (iNtRON Biotechnology, Seongnam, Republic of Korea), and Real-Time RT-PCR was performed in a Step One Real-Time PCR System (Applied Biosystems, Waltham, MA, USA). as previously reported [31]. The primers used in this study are shown in Appendix A.

### 2.11. Protein Extraction and Western Blot Analysis

Cells were harvested and then lysed in ice-cold radioimmunoprecipitation assay (RIPA) buffer. The protein concentration was determined using a protein assay reagent (Bio-Rad Laboratories Inc., Hercules, CA, USA). Western blot analysis was performed as previously described [31]. Briefly, equal amounts (20 μg) of total protein were resolved by sodium dodecyl sulfate-polyacrylamide gel electrophoresis (SDS-PAGE) (10–12% based on the size of detecting protein) and transferred to a polyvinylidene difluoride (PVDF) membrane, which were incubated with primary antibodies (overnight, 4 °C), and then incubated with the proper horseradish peroxidase-conjugated secondary antibody diluted at 1:10000 dilution (1 h, room temperature (RT)) (Jackson Immuno Research, West Grove, PA, USA).

### 2.12. Mitochondrial Microscopic Analysis and Immunofluorescence Staining

The mitochondrial analysis was performed using a Mito-Tracker Red probes CM-XRos (Invitrogen, Carlsbad, CA, USA) as previously described [30]. Fluorescence signal images were obtained using an IX71 confocal microscope (Zeiss, Oberkochen, Germany).

The immunofluorescence staining was performed as previously described [33]. Briefly, after treatment with or without CA, brown adipocytes were fixed with 4% formaldehyde and permeabilized with PBS containing 0.25% Triton X-100. After blocking non-specific binding with 5% BSA/PBS, the cells were incubated with UCP1 and PGC1α antibodies in 5% BSA/PBS (overnight, 4 °C), and incubated with fluorescent secondary antibody Alexa Fluor 488 and Alexa Fluor 546 (1 h, RT). Nuclei staining was performed by incubation with 4′,6-diamidino-2-phenylindole (DAPI). The immunofluorescence-stained images were acquired using a fluorescence microscope (Logos Biosystems, Anyang, Republic of Korea).

### 2.13. Statistical Analysis

All data were expressed as mean ± S.E.M., and processed statistically using SPSS 20 for Windows (SPSS Inc., Chicago, IL, USA). Values with *p* < 0.05 calculated by a Kruskal-Wallis H test followed by a post hoc test of Bonferroni’s method was considered statistically significant.

## 3. Results

### 3.1. Chromatographic Characterization of CA

The immature dried fruits of *Citrus aurantium* Linné were extracted by 80% ethanol. The result from HPLC chromatogram showed that CA contains two abundant compounds, naringin and neohesperidin (Figure 1). The naringin and neohesperidin was 20.132% (0.916 mg/mL) and 14.440% (0.657 mg/mL) of the 80% ethanol extract of CA 4.55 mg/mL, respectively.

### 3.2. CA Suppressed Body Weight Gain in HFD-induced Obese C57BL/6 Mice

To investigate whether CA can regulate weight of HFD-fed mice, after inducing obesity with HFD for 4 weeks, CA of 100 mg/kg/day with HFD was orally administered to C57BL/6 mice for additional 8 weeks. As shown in Figure 2A, HFD group and HFD plus CA group showed significant differences in the gain of body weight at eleventh week (24.89 ± 0.21 g vs. 15.08 ± 0.97 g). At end of the experiment, the body weights of the two groups were showed significant difference (50.31 ± 0.84 g vs. 40.91 ± 0.91 g, respectively), adipose tissue weights decreased by 35.64% in CA-treated obese mice (HFD, 0.56 ± 0.14 g vs. CA, 0.36 ± 0.08 g, p = 0.0143) (Figure 2B), and the serum total cholesterol level was significantly attenuated by CA in HFD-fed mice (Figure 2C) (*p* < 0.05).

### 3.3. CA Decreased Lipid Accumulation by Inhibiting Adipogenesis of 3T3-L1 White Adipocytes

First, to determine the concentration dependent effect of CA (10-1000 μg/mL) on cell viability, 3T3-L1 preadipocytes were applied to MTS reagent. Treatment with 10-1000 μg/mL CA for 48 h did not display any cytotoxic effect in 3T3-L1 cells (p > 0.05, Figure 3A), thus we chose concentrations of 100, 100, and 1000 μg/mL for further investigations. Next, to determine whether CA can block white adipogenesis, 3T3-L1 preadipocytes were differentiated with MDI, and the effects of CA were analyzed. As shown in Figure 3B, CA (1000 µg/mL) suppressed lipid accumulation, and this was confirmed by the reduced absorbance of Oil Red O (*p* < 0.05). At the same time, the mRNA expression involved in white adipogenesis (*Pparg*, *Cebpa*, *Fabp4*, *Adipoq* and *Retn*), which were markedly increased during MDI-induced adipogenesis, were attenuated by CA (1000 µg/mL of CA concentration; p = 0.032 for *Pparg*, p = 0.041 for *Cebpa*, p = 0.004 for *Fabp4*, p = 0.002 for *Adipoq*, and p = 0.001 for *Retn*, respectively) (Figure 3C). In addition, CA treatment reduced levels of PPARγ and C/EBPα, which are both well-known key regulators of adipocyte differentiation (Figure 3D).

### 3.4. CA Increased Differentiation of Primary Cultured Brown Adipocytes

Brown fat specializes in energy expenditure to reduce weight gain through thermogenesis, thus, we evaluated the effect of CA on thermogenesis to explore their involvement with respect to weight loss in HFD-induced obese mice. First, we assessed the cytotoxicity of CA (10-1000 μg/mL) in primary cultured brown adipocytes obtained from interscapular BAT. As in Figure 4A, no specific decrease in brown adipocyte viability was observed in treatment with 10-1000 μg/mL CA for 48 h. We next investigated whether CA treatment affected the development of primary cultured brown adipocytes by treating them with CA (10 or 1000 µg/mL) during brown adipocyte differentiation. According to our data, the lipid accumulation at a concentration of 1000 µg/mL of CA were obviously increased (*p* < 0.05) than that of the non-treated control during brown adipocyte differentiation (Figure 4B).

To explore whether the brown adipocyte function agreed to the effect of CA on differentiation, we further focused on the mitochondrial number or mass. We stained brown adipocytes with Mito-Tracker Red, which is a red fluorescent dye that specifically stains mitochondria [34], to observe significantly strong signals in the cytoplasm of CA-treated cells (p = 0.001), which indicates increased mitochondrial copy number (Figure 4C).

### 3.5. CA Induced Thermogenic Factors in Primary Cultured Brown Adipocytes

Multiple factors participate in the differentiation and thermogenic activation of BAT [17]. Our next goal was to elucidate the effect of CA on the related factors in such process. As expected, CA enhanced the mRNA levels of both thermogenic and mitochondrial specific genes, including *Pgc1a*, *Ucp1*, *Prdm16*, *Sirt3*, *Cidea*, and *CytC* (1000 µg/mL of CA concentration; *p* = 0.043 for *Pgc1a*, *p* = 0.050 for *Ucp1*, *p* = 0.01 for *Prdm16*, *p* = 0.050 for *Sirt3*, *p* = 0.021 for *Cidea*, and *p* = 0.012 for *CytC*) (Figure 5A). We also analyzed protein levels of UCP1 and PGC1α, which were significantly increased by CA at 1000 µg/mL compared to non-treated control during differentiation (Figure 5B), demonstrating the thermogenic effects of CA. After observing an upregulation of UCP1 and PGC1α, we further examined the expression levels of those by using immunofluorescence assay. We observed that the fluorescent protein levels of UCP1 and PGC1α were markedly induced by CA treatment (1000 µg/mL) during brown adipocytes differentiation (p = 0.001 for PGC1α and p = 0.043 for UCP1) (Figure 5C).

The recruitment of functional brown adipocytes, also called beige or brite adipocytes, is another rising target for obesity treatment [35]. However, in contrast to results from brown adipocytes, CA failed to induce the thermogenic potential in mature white adipocytes. There was no significant up-regulation in both UCP1 and PGC1α by CA treatment in 3T3-L1 adipocytes (Appendix A).

### 3.6. CA Regulated Adipogenesis and Thermogenesis via AMPK Activation

AMPK, the energy sensor and also the master regulator of metabolic homeostasis, acts to increase glucose uptake, fatty acid oxidation, and mitochondrial biogenesis [36]. In addition, AMPK is involved in both thermogenic activation of BAT and browning of WAT as well as conversion of preadipocytes to mature fat cells [37]. Based on these reports, to elucidate whether the effect of CA on promoting adipogenesis and thermogenesis was dependent on the activation of AMPK, 3T3-L1 adipocytes and primary adipocytes were treated with CA at indicated concentration during differentiation. As a result, CA (1000 µg/mL) in both 3T3-L1 cells and primary brown adipocytes enhanced the levels of pLKB1, the upstream signal of AMPK, pAMPKα, and pACC, which is phosphorylated by AMPK [38]. without significant changes in total AMPK (*p* < 0.001 for pAMPKα and p = 0.001 for pACC) (Figure 6A). Phosphorylated AMPKα by CA was successfully blocked after an AMPK inhibitor CC treatment in the both cells (Figure 6B,C). In addition, perturbed induction of PPARγ and C/EBPα by CA during the differentiation was reversed by CC treatment in 3T3-L1 cells. Concurrently, the CA-induced PGC1α and UCP1 expression were attenuated by CC treatment in primary brown adipocytes (Figure 6D,E). These results denote that adipocytes with AMPK deficiency is specifically affected in the expression of adipogenesis- and thermogenesis-related genes and proteins in both 3T3-L1 adipocytes and primary cultured brown adipocytes.

## 4. Discussion

Since obesity is a common chronic disease that results from dysregulated adipogenesis caused by imbalance of food intake and EE, therapeutic strategies targeting regulation of adipogenic differentiation or increase EE are both attractive approaches for combating obesity [39,40]. Generally, classic brown adipocytes increase EE by producing heat, which protects against the development of excess lipid accumulation, and eventually prevents overweight and obesity. Approaches to elevate the thermogenic activity of brown and beige adipocytes are potentially promising new strategies for preventing and treating obesity because the activity of these thermogenic adipocytes is inversely correlated with fat mass and positively related to EE [41].

Medications for obesity treatment vary, but only five are currently approved by the Food and Drug Administration (FDA) of USA [42]. However, the search for alternative treatments is still an ongoing challenge due to the various side effects of current medications. In context, some natural compounds such as polyphenol and flavonoids offer an alternative therapeutic strategy, as they can reduce the accumulation of lipid by inhibiting lipogenesis or inducing lipolysis, and subsequently decrease adiposity [43], or induce EE by increasing UCP1 in BAT [30,33]. However, there has been little evidence that CA can directly affect adipogenesis of white adipocytes and thermogenesis of brown adipocytes which are both attractive targets of anti-obese strategy. The current study evaluated the effect of CA on adipocyte differentiation in WAT and thermogenic capacity of BAT in both 3T3-L1 white adipocytes and primary cultured brown adipocytes. Our findings revealed that administration of CA significantly reduced body weight in obese mice, suggesting that CA may be potential option as an intervention to fight obesity.

Adipogenesis is a complex process of which several factors involve. Among these factors, PPARγ and C/EBPα are considered as the most crucial regulators of adipogenesis [11]. They mutually induce expressions, cooperate as transcription factors, and eventually leads to synergistic activation of other adipocyte factors [12]. In our study, yet the effect of CA on lipid accumulation may not seemed to be impressive (17.6% decrease at 1000 μg/mL concentration), but the adipogenesis regulators, PPARγ and C/EBPα, were suppressed by 21.6% and 34.6% respectively by CA treatment (1000 μg/mL), indicating a potentially strong effect on the adipogenesis cascade. The results indicated that the CA-reduced lipid accumulation is attributed to a decrease in these crucial factors associated with white adipogenesis. In addition, the genes those encode adipokines such as adipocyte protein 2 (aP2), adiponectin and resistin, were significantly suppressed by CA treatment, indicating reduced adipocyte development.

Considering the benefits and safety of BAT-mediated EE, this unique tissue can be the core of next generation for anti-obesity agents. Several factors are related to the process of non-shivering thermogenesis [44]. UCP1 in the inner membrane of mitochondria is preferably abundant in brown adipocytes. By uncoupling oxidative phosphorylation from ATP production, UCP1 converts triglyceride to heat [45]. The PR-domain containing 16 (PRDM16), transcribed from the gene *Prdm16*, is the inducer of the thermogenic phenotype, and activates the brown fat program in adipocytes via *Pgc1a*, *Cidea* and *CytC* [45,46]. On the other hand, sirtuin 3 is a member of the sirtuin family of protein deacetylases which preferentially localize in mitochondria [47,48], and is recently recognized as an important regulator of non-shivering thermogenesis program [49]. We found that CA promotes activation of BAT function and differentiation by upregulation of genes related to BAT thermogenesis, mitochondrial biogenesis and mitochondrial activation such as *Pgc1a*, *Ucp1*, *Prdm16*, *Sirt3*, *Cidea* and *CytC*, thereby suppress obesity by promoting EE. However, in the concept of ‘browning’, a rising concept of approach in obesity care of which strategy is aimed to recruit functional brown adipocytes within WAT [50], CA failed to alter any thermogenic factors including UCP1 and PGC1α. This allowed us to conclude CA may benefit obesity by suppressing adipogenesis of white adipocytes but not inducing ‘browning’ of it. The thermogenic action of CA may only happen in BAT, yet it is known that UCP1 activation in BAT can systemically affect lipolysis of WAT to produce fatty acid for thermogenic fuel supply [51].

According to previous reports, CA displays potentially beneficial effects on metabolic diseases such as obesity and type 2 diabetes [25,26,27,28]. However, the underlying mechanism of the anti-obese effect of CA remained obscure so far. In this study, based on the crucial role of AMPK in metabolic diseases, we attempted to reveal the effect of CA on adipogenesis and thermogenesis and assess the role of AMPK in this process. AMPK is a serine/threonine kinase of a heterodimer complex comprising a catalytic α subunit (α1, α2) and two regulatory subunits β (β1, β2) and γ (γ1, γ2, γ3). When AMPKα is phosphorylated, ATP consumption (anabolism) is inhibited and ATP production (catabolism) are activated. The overall effect of AMPKα activation is therefore to synthetize ATP and restore AMP:ATP and ADP:ATP ratios for the maintenance of cellular energy homeostasis [52,53,54]. AMPK acts as a regulator of white adipogenesis, brown adipogenesis, and brown adipocyte activation. The different role of AMPK in these three processes has been established previously. In white adipocyte differentiation (or adipogenesis), AMPK negatively regulates this process. Treatment of AICAR, the AMPK activator, in 3T3-L1 or F442A preadipocytes led to inhibition of differentiation accompanied by decreased PPARγ and C/EBPα [37,55]. On the other hand, in brown adipocytes, AMPK is essential for their development. Deletion of AMPK β subunits or AMPKα1 in mice resulted in defected brown adipocyte development [56,57]. Moreover, AMPK plays a crucial role in the activation of non-shivering thermogenesis in brown fat as well [58,59]. Thus, we attempted to identify whether AMPK pathway is involved in the action of CA. In consistent with previous reports on AMPK, we could observe that CA elevated phosphorylation levels of AMPKα and ACC both in 3T3-L1 adipocytes and primary cultured brown adipocytes. Furthermore, inhibition of AMPKα pathway by pre-treatment of CC lead to abolishment of the anti-adipogenic and pro-thermogenic effect of CA. Thus, we could assume the anti-obese effect of CA was dependent on the activation of AMPKα pathway.

Hit identification is considered as the basic level of small molecule discovery. Thus, it is necessary to identify effective constituents which can represent the effects of natural products. CA contains many compounds, including p-Octopamine and synephrine alkaloids. [26] The flavonoids, also isolated from CA, of which hesperidin and naringenin are major active compounds have been used to treat cardiovascular diseases [60]. Through and HPLC analysis, we identified two constituents which may possibly be responsible for the effect of CA. Naringin is a component derived from most citrus fruits including grapefruit and CA. Recent studies by Pu et al. [61] and Sui et al. [62] has shown that naringin can activate AMPK and thereby reduced body weight in HFD-fed obese C57BL/6J mice. Neohesperidin, another well-known compound of the Citrus genus, also has an AMPK-dependent lipid-regulating effect in vivo and vitro [63]. Further studies reported the anti-adipogenic or anti-obese effect of naringin [64,65,66,67] and neohesperidin [68,69], nevertheless, none of these have established the effect of either compound on UCP1-mediated thermogenesis. Our LC-MS data suggests that these two constituents may be responsible for the AMPKα-dependent anti-adipogenic and thermogenic effect of CA, however, to clarify, further investigation should be carried out with different fractions and/or identified constituents.

## 5. Conclusions

In conclusion, our study revealed the underlying mechanism of the anti-adipogenic effect of CA, of which AMPKα acts as a crucial factor. Moreover, a new therapeutic possibility of CA; induction of non-shivering thermogenesis, was suggested by our study regarding primary cultured brown adipocytes. The activation of brown adipocytes was also dependent on AMPKα as well. Overall, we suggest CA as a new potential anti-obese agent which can inhibit white adipogenesis and induce brown adipocyte thermogenesis via activation of AMPKα.

## Figures and Tables

**Figure 1 nutrients-11-01988-f001:**
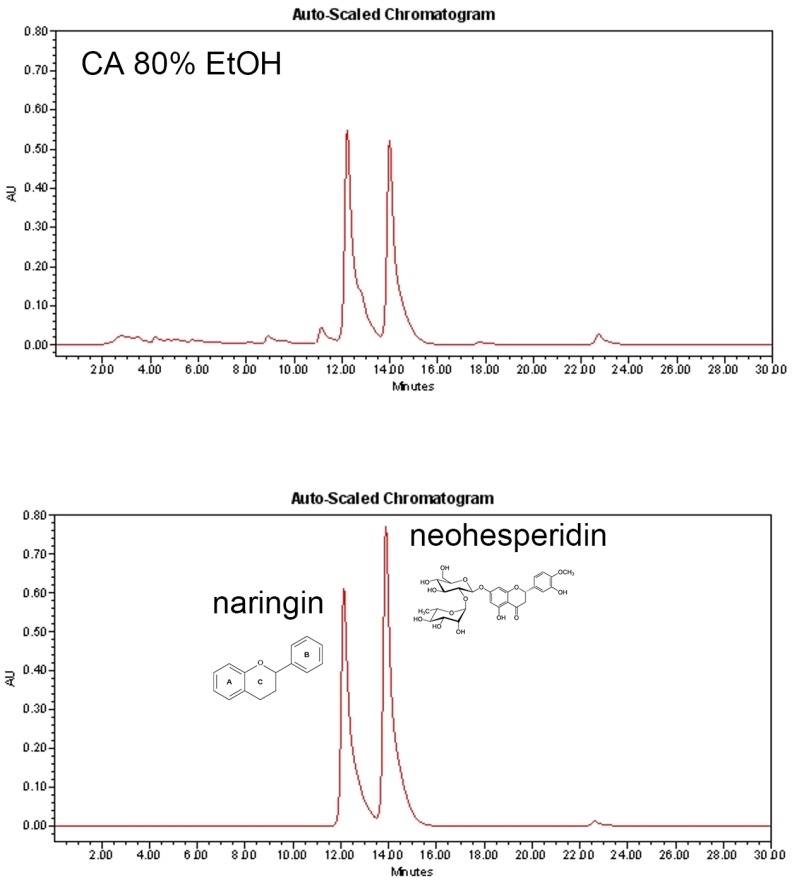
HPLC chromatogram of CA. Each peak displays CA (above) and naringin and neohesperidin (below). CA was extracted with 80% ethanol. CA, bitter orange (*Citrus aurantium* Linné).

**Figure 2 nutrients-11-01988-f002:**
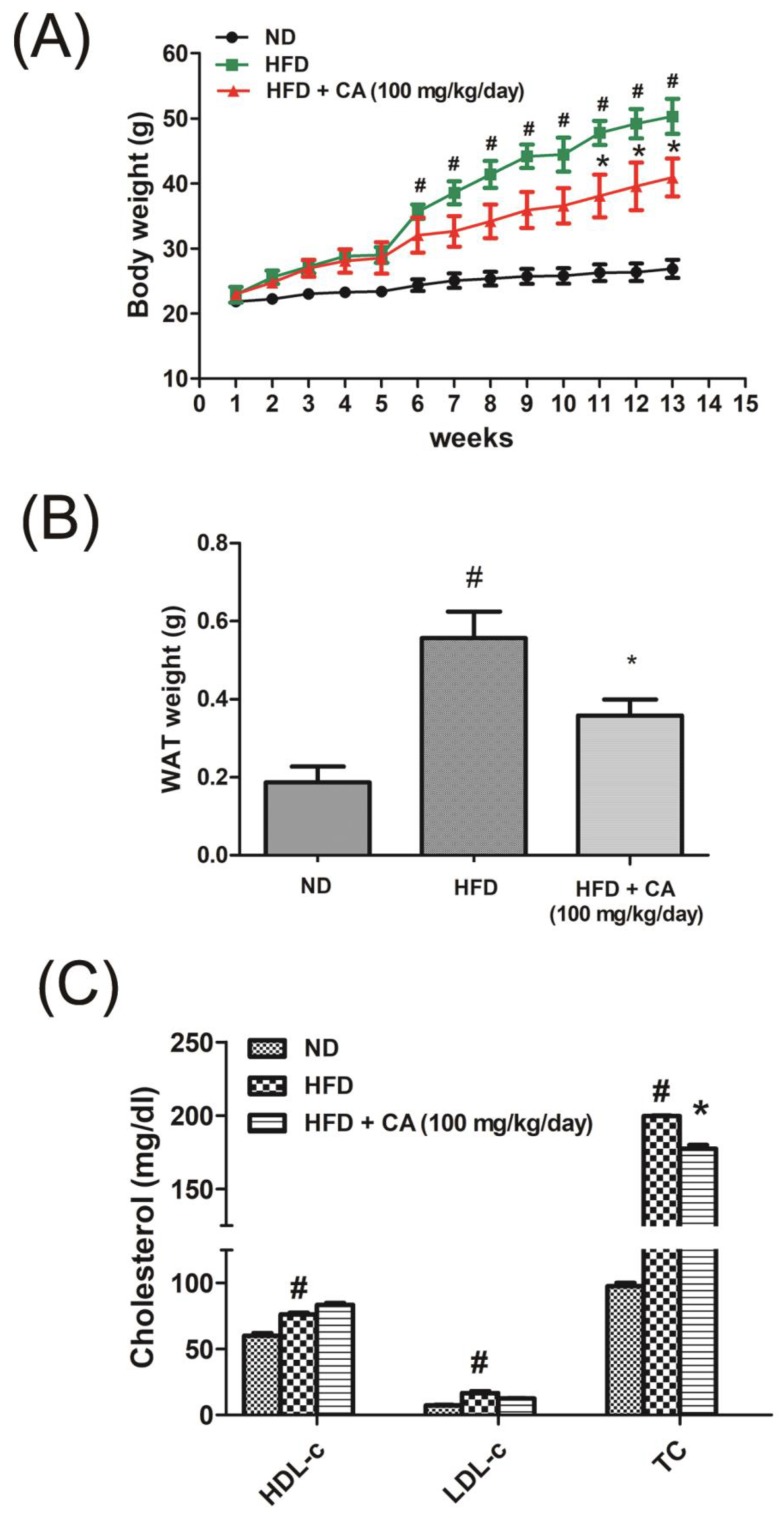
Effect of CA on weight gain in HFD-induced obese mice and on adipogenic differentiation. (**A**) Body weight and (**B**) adipose tissue weight changes of C57BL/6J mice. (**C**) Serum lipid profile of C57BL/6J mice were measured. ^#^ p < 0.05 vs. the ND group, * p < 0.05 vs. the HFD group. ND, normal diet-fed control group; HFD, high fat diet-fed obese group; CA, high fat diet supplemented with 100 mg/kg/day bitter orange (*Citrus aurantium* Linné) -fed obese group.

**Figure 3 nutrients-11-01988-f003:**
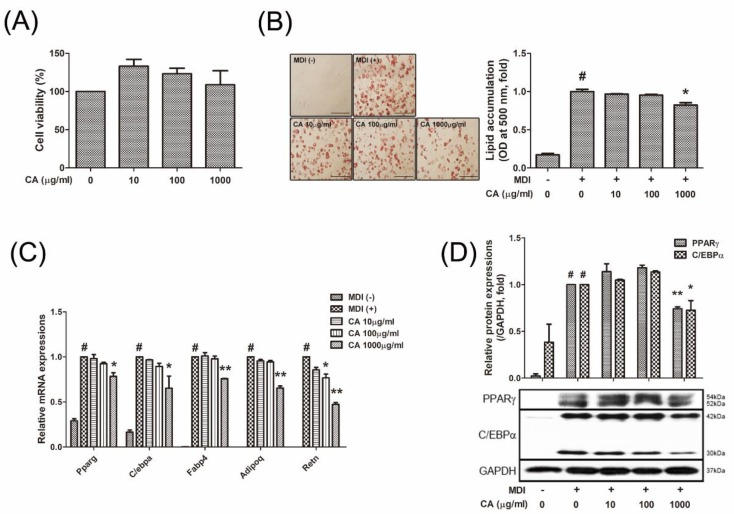
Effect of CA on adipogenesis in 3T3-L1 adipocytes. (**A**) Cytotoxicity of CA in 3T3-L1 preadipocytes were determined by an MTS assay. (**B**) Lipid accumulation was measured by Oil Red O staining (magnification 200×, scale bar 100 µm). (**C**) The mRNA level of *Pparg, Cebpa, Fabp4, Adipoq* and *Retn* were analyzed by Real-Time RT-PCR. (**D**) PPARγ and C/EBPα protein levels were analyzed by a western blot analysis. Quantification of the protein bands was measured using Image J. Results were expressed relative to GAPDH. All data are presented as the mean ± SEM. ^#^*p* < 0.05 versus the MDI-uninduced preadipocytes, * *p* < 0.05 versus the MDI-induced adipocytes, ** *p* < 0.01 vs. the MDI-induced adipocytes. MDI, differentiation medium; CA, bitter orange (*Citrus aurantium* Linné).

**Figure 4 nutrients-11-01988-f004:**
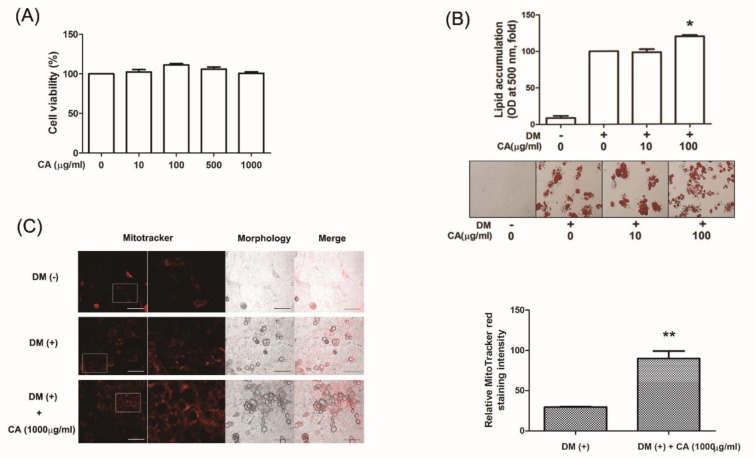
Effect of CA on development of primary cultured brown adipocytes. (**A**) Cytotoxicity of CA in primary cultured brown preadipocytes were determined by an MTS assay. (**B**) Lipid accumulation was measured by Oil Red O staining (magnification 200×, scale bar 100 µm). (**C**) Mitochondrial abundance in primary cultured brown adipocytes was analyzed by MitoTracker Red staining. Quantification of the Mitotracker signaling was measured using Image J from five separate slides. * *p* < 0.05 vs. the DM-induced adipocytes, ** *p* < 0.01 vs. the DM-induced adipocytes. DM, differentiation medium; CA, bitter orange (*Citrus aurantium* Linné).

**Figure 5 nutrients-11-01988-f005:**
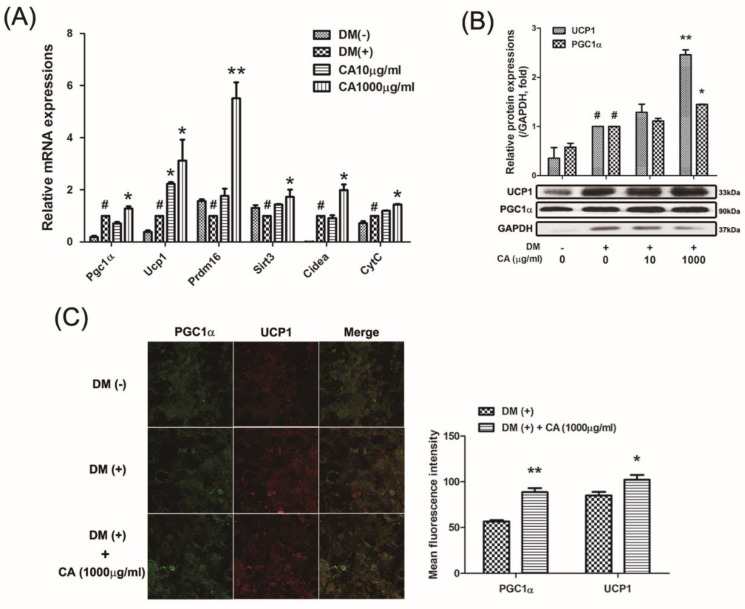
Effect of CA on thermogenic factors in primary cultured brown adipocytes. (**A**) The mRNA level of *Pgc1a, Ucp1, Prdm16, Sirt3, Cidea* and *CytC* were analyzed by Real-Time RT-PCR. (**B**) UCP1 and PGC1α protein levels were analyzed by a western blot analysis. Quantification of the protein bands was measured using Image J. Results were expressed relative to GAPDH. (**C**) Immunofluorescence staining of UCP1 (red) and PGC1α (green) were performed (magnification 200×, scale bar 100 µm). Quantification of the immunofluorescence signaling was measured using Image J from five separate slides. All values are means ± SEM of data from three separate experiments. ^#^*p* < 0.05 versus DM-uninduced preadipocytes; and ^*^*p* < 0.05 and ^**^*p* < 0.01 versus DM-induced adipocytes. DM, differentiation medium; CA, bitter orange (*Citrus aurantium* Linné).

**Figure 6 nutrients-11-01988-f006:**
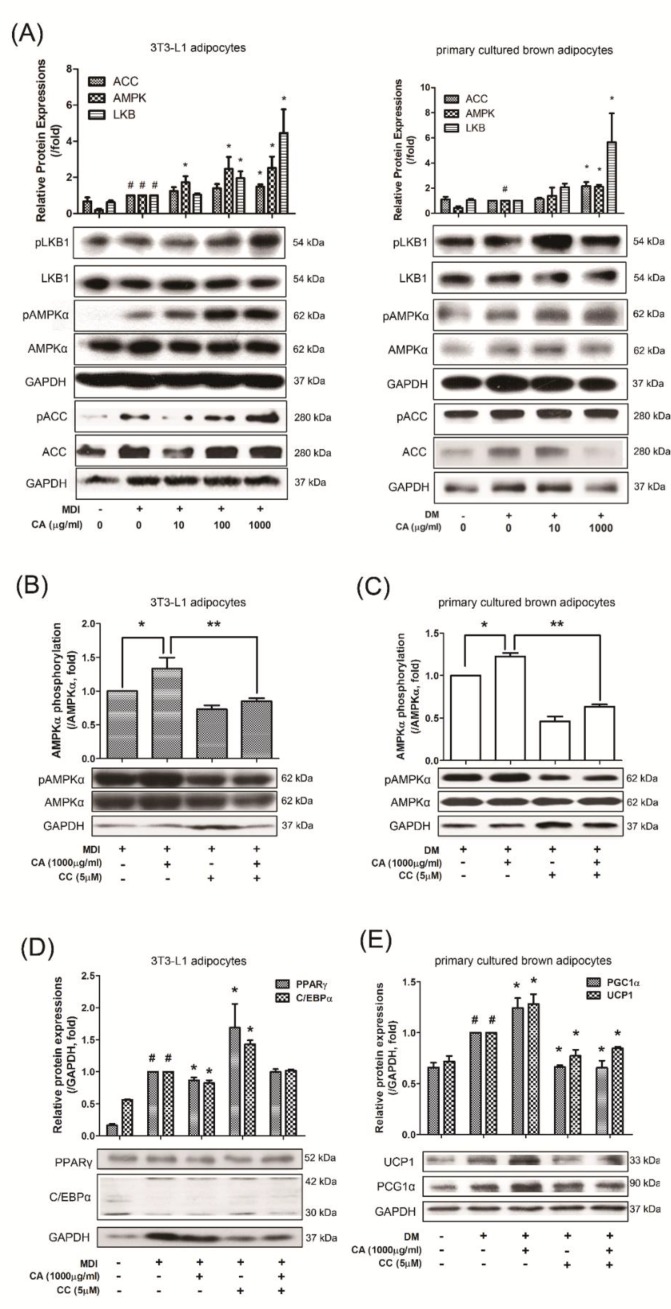
Effects of CA on activation of AMPKα pathway and in AMPKα-inhibited conditions of 3T3-L1 cells and primary cultured brown adipocytes. (**A**) Phosphorylation levels of AMPKα, LKB1 and ACC were determined in 3T3-L1 adipocytes and primary cultured brown adipocytes. pAMPKα, LKB1 and pACC protein expressions were normalized to total AMPKα, LKB1 and ACC, respectively. (**B, C**) Inhibition of phosphorylation of AMPKα by CC co-treatment was determined by Western blot analysis. (**D**, **E**) PPARγ, C/EBPα, PGC1α, and UCP1 expressions under co-treatment of CA and CC were analyzed by western blot analysis. Quantification of the protein bands was measured using Image J. Results were expressed relative to GAPDH. All values are means ± SEM of data from three separate experiments. ^#^*p* < 0.05 versus MDI- or DM-uninduced preadipocytes; and ^*^*p* < 0.05 and ^**^*p* < 0.01 versus the MDI- or DM-induced adipocytes. MDI and DM, differentiation medium; CA, bitter orange (*Citrus aurantium* Linné); CC, compound C.

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
