# Peer review of "Bitter Orange (Citrus aurantium Linné) Improves Obesity by Regulating Adipogenesis and Thermogenesis through AMPK Activation"

_nutrients, 2019, doi:10.3390/nu11091988_

Round 1
Reviewer 1 Report
The authors investigated that the molecular mechanism of anti-obese effects of bitter orange (Citrus aurantium Linné, CA) in high fat diet-fed mice. Administration of CA lowered body weight and improved hypercholesterolemia. CA suppressed adipocyte differentiation of 3T3-L1 cells, but its suppression effect is very subtle. Interestingly, CA induced the expression of thermogenesis-associated genes in primary culture brown adipocyte. CA-mediated suppression of adipogenesis occurs through activating AMPK activation. The results are sound. However, there are concerns that should be addressed.
1. What amount of proteins were loaded in each lane in Western blot analysis?
2.Ingredients of the two diets, LFD and HFD such as proteins, fibers, etc. are comparable?
3.The reason for choosing CA concentration of 100 mg/kg should be addressed.
4.The body weight gain was clearly reduced by administration of CA to HFD-fed mice (Fig. 2A). However, the reduction of intracellular lipid level in 3T3-L1 cells was a little in in vitro analysis. The reduction of body weight gain is derived from a decrease in activation of brown adipose tissue? The authors should explain what the difference between in vivo and in vitro is.
5.In the differentiated 3T3-L1 cells, was the expression level of UCP-1 measured, when CA was administered? CA induces browning of 3T3-L1 cells?
\
Author Response
The authors investigated that the molecular mechanism of anti-obese effects of bitter orange (Citrus aurantium Linné, CA) in high fat diet-fed mice. Administration of CA lowered body weight and improved hypercholesterolemia. CA suppressed adipocyte differentiation of 3T3-L1 cells, but its suppression effect is very subtle. Interestingly, CA induced the expression of thermogenesis-associated genes in primary culture brown adipocyte. CA-mediated suppression of adipogenesis occurs through activating AMPK activation. The results are sound. However, there are concerns that should be addressed.
1. What amount of proteins were loaded in each lane in Western blot analysis?
ANSWER: 20 μg of protein was loaded for western blot analysis. This information has been added in the Materials and Methods section (page 4, line 151).
2.Ingredients of the two diets, LFD and HFD such as proteins, fibers, etc. are comparable?
ANSWER: We included the ingredient compositions of NC and HFD as Table 1. Thank you.
3.The reason for choosing CA concentration of 100 mg/kg should be addressed.
ANSWER: The administration dose of CA was decided based on a previous study [Biol Pharm Bull. 2019;42(2):255-260.]. Although the disease models differ (NFALD vs. obesity), the method of disease induction (HFD administration) was identical, thus we decided 100 mg/kg/day as the proper dose of CA for our study. This information has been added in the Materials and Methods section (page 3, line 105-106).
4.The body weight gain was clearly reduced by administration of CA to HFD-fed mice (Fig. 2A). However, the reduction of intracellular lipid level in 3T3-L1 cells was a little in in vitro analysis. The reduction of body weight gain is derived from a decrease in activation of brown adipose tissue? The authors should explain what the difference between in vivo and in vitro is.
ANSWER: Our results suggest a decrease in adipogenesis of WAT and activation of BAT-mediated thermogenesis is the two mechanisms responsible for the weight loss in HFD-fed mice. As the reviewer pointed out, yes, the effect of CA on lipid accumulation (Fig 3B) may not seem impressive (17.6% decrease at 1000 μg/ml concentration), but the adipogenic factors such as PPARγ and C/EBPα were suppressed by 21.6% and 34.6% respectively by CA treatment (1000 μg/ml). These two results suggest CA might not be effective enough to inhibit adipogenesis at a sufficient level, but the reduced adipogenic factors, probably a subsequent phenomenon of AMPK activation induced by CA, leads to at least a moderate decrease in adipocyte differentiation. Furthermore, our newly included data on adipose tissue weight suggests CA significantly reduced the weight of WAT (HFD group 0.557 g vs. CA-treated group 0.3585 g, p = 0.0143) (Figure 2B). We suppose this effect was due to the activation of BAT, as reports show that UCP1 activation in BAT systemically induces lipolysis in WAT to use fatty acid as thermogenic fuel [Cell Metab. 2017 Nov 7;26(5):764-777.]. As CA failed to induce browning of white adipocytes (included as Supplementary Figure S1 in revised manuscript), our results overall suggests that CA can activate AMPK in both brown and white adipocytes/adipose tissues and this leads to inhibition of adipogenesis in WAT and activation of thermogenesis in BAT. Relevant information has been included in the Discussion section (page 13, line 324-327; page 14, line 343-349, 359-368).
5.In the differentiated 3T3-L1 cells, was the expression level of UCP-1 measured, when CA was administered? CA induces browning of 3T3-L1 cells?
ANSWER: We are aware of the fact that browning, or trans-differentiation of white adipocytes to functional thermogenic adipocytes, is an uprising target for obesity care. As the reviewer suggested, we evaluated the browning effect of CA in 3T3-L1 adipocytes. Unexpectedly, CA failed to induce UCP1 or PGC1α upregulation. Thus, we conclude CA induces the thermogenic program only in brown adipocytes/BAT but not in white adipocytes/WAT. Relevant information has been included in the Results section as Supplementary Figure S1 as attached (p. 10, line 257-261).

Reviewer 2 Report
In this study, the authors aimed to examine the anti-adipogenic and thermogenic mechanisms of Citrus aurantium Linné (CA) in 3T3-L1 adipocytes and primary brown adipocytes, focusing on the role of AMP-activated protein kinase (AMPK). The authors found that their results revealed the underlying mechanism of the anti-adipogenic effect of CA, of which AMPKα acts as a crucial factor, and an induction of non-shivering thermogenesis in primary cultured brown adipocytes. The authors concluded that CA could be as a new potential anti-obese agent, which inhibited white adipogenesis and induce brown adipocyte thermogenesis via activation of AMPKα.
Comments
This is an interesting study. However, the reviewer has some major concerns as follows:
1. How about the weight of adipose tissue? It is the important data for anti-obese and anti-adipogenic effects of CA.
2. CA at high concentration inhibited the lipid accumulation in 3T3-L1 adipocytes, but increased the lipid accumulation in primary cultured brown adipocytes. Why CA induced the inverse effects on lipid accumulation in these two kinds of cells, but increased AMPK phosphorylation in these two kinds of cells? Does AMPK regulate both decrease and increased in lipid accumulation?
3. In Figure 4C, the images for Mitotraker staining are not convincing. The stain intensities are week in DM(+) and DM(+)+CA. The Mitotraker staining and cell morphology is really not matched.
4. In Figure 5C, the images for immunofluorescence staining of UCP1 (red) and PGC1α (green) are also not convincing. It is hard to distinguish the difference between DM and DM+CA groups that the data for the increase in UCP1 and PGC1α are suspect.
5. In Figure 6A, the expression of pAMPK in MDI group in 3T3-L1 adipocytes is not increased as compared to control group. The statistical analysis should be confirmed. Moreover, why the total form of AMPKα was markedly decreased in DM+CA (1000 μg/ml) group?
Author Response
In this study, the authors aimed to examine the anti-adipogenic and thermogenic mechanisms of Citrus aurantium Linné (CA) in 3T3-L1 adipocytes and primary brown adipocytes, focusing on the role of AMP-activated protein kinase (AMPK). The authors found that their results revealed the underlying mechanism of the anti-adipogenic effect of CA, of which AMPKα acts as a crucial factor, and an induction of non-shivering thermogenesis in primary cultured brown adipocytes. The authors concluded that CA could be as a new potential anti-obese agent, which inhibited white adipogenesis and induce brown adipocyte thermogenesis via activation of AMPKα.
Comments
This is an interesting study. However, the reviewer has some major concerns as follows:
1. How about the weight of adipose tissue? It is the important data for anti-obese and anti-adipogenic effects of CA.
ANSWER: As the reviewer suggested, we have included the tissue weight of WAT in Figure 1 as attached.
2. CA at high concentration inhibited the lipid accumulation in 3T3-L1 adipocytes, but increased the lipid accumulation in primary cultured brown adipocytes. Why CA induced the inverse effects on lipid accumulation in these two kinds of cells, but increased AMPK phosphorylation in these two kinds of cells? Does AMPK regulate both decrease and increased in lipid accumulation?
ANSWER: AMPK acts as a regulator of white adipogenesis, brown adipogenesis, and brown adipocyte activation. The different role of AMPK in these three processes has been well-established. First, in white adipocyte differentiation (or adipogenesis), AMPK negatively regulates this process. Treatment of AICAR, an activator of AMPK, in 3T3-L1 or F442A preadipocytes led to inhibition of differentiation accompanied by inhibited PPARγ and C/EBPα [Biochem Biophys Res Commun. 2001 Sep 7; 286(5):852-6.; Biochem Biophys Res Commun. 2006 Feb 3; 340(1):43-7.]. On the other hand, in brown adipocytes, AMPK is essential for their development. Deletion of AMPK gene results in defected brown adipocyte development [Biochem Biophys Res Commun. 2017 Sep 16;491(2):508-514.; Cell Metab. 2016 Jul 12;24(1):118-29.], and moreover, AMPK plays a crucial role in the activation of non-shivering thermogenesis in brown fat as well [Diabetologia. 2005 Nov;48(11):2386-95.; Diabetes. 2014 Oct;63(10):3346-58.]. In consistent with previous reports on AMPK, our results demonstrated the AMPK-mediated dual effect of CA on two different types of adipocytes: The Oil Red O staining in Figure 1 is performed in 3T3-L1 white adipocytes, and the one shown in Figure 4 is performed in primary cultured brown adipocytes. CA inhibited adipogenesis of white adipocytes and induced development of brown adipocytes. Relevant information has been included in the Discussion section (page 14, line 359-368).
3. In Figure 4C, the images for Mitotraker staining are not convincing. The stain intensities are week in DM(+) and DM(+)+CA. The Mitotraker staining and cell morphology is really not matched.
ANSWER: We have included a figure with better resolution to avoid misunderstandings. The cell morphology and MitoTracker staining were captured from identical slides. The fluorescence signals in Figure 4C were measured by Image J from 5 or more slides which were randomly chosen by a blinded researcher. The mean ± S.E.M. of each group was as following: DM, 29.59 ± 0.49; DM+CA, 90.04 ± 9.23.
4. In Figure 5C, the images for immunofluorescence staining of UCP1 (red) and PGC1α (green) are also not convincing. It is hard to distinguish the difference between DM and DM+CA groups that the data for the increase in UCP1 and PGC1α are suspect.
ANSWER: We have included a figure with better resolution to avoid misunderstandings. The fluorescence signals in Figure 5C were measured by Image J from 5 or more slides which were randomly chosen by a blinded researcher. The mean ± S.E.M. of each group was as following: DM, 85.00 ± 3.97 (UCP1) and 56.62 ± 1.49 (PCG1α); DM+CA, 102.35 ± 5.10 (UCP1) and 88.73 ± 4.35 (PGC1α).
5. In Figure 6A, the expression of pAMPK in MDI group in 3T3-L1 adipocytes is not increased as compared to control group. The statistical analysis should be confirmed. Moreover, why the total form of AMPKα was markedly decreased in DM+CA (1000 μg/ml) group?
ANSWER: We have performed additional western blot analysis to confirm this matter. To answer the reviewer’s concerns, further assays showed that phosphorylation of AMPK was markedly increased by differentiation media (Figure 6A). The total AMPK in brown adipocytes seems to be decreased, however the blot signals of tAMPK turned out to be quite stable when confirmed by Image J evaluation. We believe the decrease was caused from weak signals. We have replaced these blots in the revised Figure 6 to avoid misunderstandings.
We also confirmed the phosphorylation of LKB1, the upstream cascade of AMPK, to be sure whether CA is able to induce AMPK activation. This result has been included in the revised Figure 6.

Round 2
Reviewer 1 Report
The manuscript was improved. I have no more comment.
Reviewer 2 Report
No further comments. This revised manuscript can be accepted to publish in this Journal.